# Evidence of Asexual Overwintering of *Melampsora paradoxa* and Mapping of Stem Rust Host Resistance in *Salix*

**DOI:** 10.3390/plants11182385

**Published:** 2022-09-13

**Authors:** Chase R. Crowell, Dustin G. Wilkerson, Lawrence B. Smart, Christine D. Smart

**Affiliations:** 1Plant Pathology & Plant-Microbe Biology Section, School of Integrative Plant Science, Cornell AgriTech, Cornell University, Geneva, NY 14456, USA; 2Horticulture Section, School of Integrative Plant Science, Cornell AgriTech, Cornell University, Geneva, NY 14456, USA

**Keywords:** *Melampsora americana*, *Melampsora paradoxa*, *Salix purpurea*, shrub willow, willow leaf rust, asexual overwintering, host resistance, population biology

## Abstract

*Melampsora* rust is a devastating disease of shrub willow in North America. Previous work has identified *Melampsora paradoxa* as one of two identified rust species in New York State that infect *Salix purpurea* and other important *Salix* host species, however little is known about the population of this rust species in this region. Genotyping-by-sequencing was used to identify single nucleotide polymorphisms (SNPs) and assess population diversity of *M. paradoxa* isolates collected from three *Salix* breeding populations in Geneva, NY between 2015 and 2020. Statistical analyses of SNP revealed that all isolates collected were clonally derived even though they were collected across years. In 2020, isolates were collected from stem infections where uredospore pustules were observed, and these isolates were also identical to *M. paradoxa* collected in previous seasons. These data suggest that *M. paradoxa* sampled across multiple years overwintered and reproduced asexually and that stem infection is a possible mechanism for overwintering, both of which are novel findings for this rust species. Additionally, field disease ratings were conducted on a *S. purpurea* × *S. suchowensis* F_1_ breeding population with high disease severity, enabling the discovery of QTL for resistance on chromosomes 1 and 19. Lastly, *Colletotrichum salicis* was frequently associated with stem rust and may play a role in *M. paradoxa* stem infection. Together, this work is the first substantial exploration into *M. paradoxa* population biology, stem infection, and host resistance in *Salix*.

## 1. Introduction

Bioenergy crops in the family Salicaceae, including *Salix* and *Populus*, are well suited for woody biomass production in much of the Northern Hemisphere [1,2,3]. Shrub willow in particular has been identified as an ideal bioenergy crop for much of the Greater Lakes Region due to its ability to thrive on wet, marginal lands [4,5]. However, *Melampsora* spp. rusts are a substantial threat to sustained vigor and yield in this region due to their ability to colonize leaf tissue leading to defoliation, potential for long distance inoculum spread via wind, and prolific asexual reproduction [3,6]. While these pathogens are quite understudied in North America, species level surveys have been performed to identify the *Melampsora* species infecting *Salix* hosts in New York (NY) and the Great Lakes Region of the US [6,7].

Kenaley et al. [7] identified three phylotaxa of *Melampsora* rust collected in New York using internal transcribed spacer region (ITS) sequences, two of which were identified as *Melampsora americana* and *Melampsora paradoxa* alternating on *Abies balsamea* and *Larix* spp. respectively. Isolates in Phylotaxon II identified as *M. paradoxa* were collected from hosts *S. purpurea*, *S. nigra*, *S. amygdaloides*, *S eriocephala*, and *S. miyabeana*. Isolates in Phylotaxon III identified as *M. americana* were collected from hosts *S. purpurea*, *S. discolor*, *S. interior*, *S. eriocephala*, *S. caprea*, *S. dasyclados* and *S. integra* hybrids. While many of these hosts species are overlapping, *M. americana* was more frequently observed on North American native willows. Crowell et al. [6] isolated the same two rust species from *S. purpurea*, however *M. americana* was collected more frequently. Several investigations have focused on *M. americana* as the primary willow leaf rust pathogen in the Northeast US [8,9], but *M. paradoxa* remains quite understudied. *Melampsora paradoxa* is macrocyclic and heteroecious and is presently the only *Melampsora* rust known to alternate on *Larix* spp. hosts in the northeast US [7]. Work conducted by Crowell et al. [6] identified *M. paradoxa* as a small subset of a large collection of isolates, all of which were collected early in the growing season. Additionally, ITS sequences were identical for the isolates collected, a characteristic not observed for *M. americana* isolates from the northeast US [6].

The goal of this study was to explore the population diversity of *M. paradoxa* isolates collected from 2015 through 2020 in three neighboring shrub willow fields using single nucleotide polymorphisms (SNP) identified across the *M. paradoxa* genome using genotyping-by-sequencing (GBS) [10] while also exploring host resistance to this *Melampsora* species. A better understanding of the *M. paradoxa* population in the northeast US provides guidance for future breeding for resistance in high yielding shrub willow cultivars, as well as a greater understanding of rust epidemics within each growing season.

## 2. Results

### 2.1. *Melampsora paradoxa* Isolate Collection

In order to better characterize the pathogen threats to willow biomass crops, a total of 61 *M. paradoxa* isolates were collected between the years 2015 to 2020 across three *Salix* breeding populations: the 2013 Family Selection Trial, a *Salix purpurea* F_2_ mapping population [11], and the *Salix* F_1_ hybrid common parent population (HCP) [12] within close vicinity of each other located at the Crittenden North Research Farm at Cornell AgriTech in Geneva NY (Table 1). In 2015, six isolates were collected from the border row of the 2013 Family Selection Trial from one host genotype. *Melampsora paradoxa* was determined as the species of isolates collected in 2015 through ITS phylogeny (Appendix A) as previously described [8]. In 2016 four *M. paradoxa* isolates were collected, one from the same plot in the border row of the collection in 2015 and the remaining three others from other host genotypes in immediate proximity on the border row. In 2017, 10 rust isolates were collected, four of which were from the same plot as from 2015, five were from other plots in the border row, and one from the *Salix* F_1_ HCP. All rust collected in the family selection trial between 2015 and 2017 were isolated from *S. purpurea* × *S. suchowensis* progeny. A single isolate from 2017 was collected from the neighboring F_2_ mapping population [11]. The first three isolates collected from 2018 were inside the first and second replicate blocks of the family selection trial (two of which were on a *S. miyabeana* host and the other on a *S. purpurea* × *S. suchowensis* hybrid progeny) and the remaining 26 were collected across the *Salix* F_1_ HCP from various *S. purpurea* hybrids. Seven isolates were collected in the year 2020. Four isolates were collected from the border row of the 2013 Family Selection Trial and one isolate was collected from sporulating cankers observed on the stem of the willow host. Two isolates were collected from dormant stem cankers that were incubated in the greenhouse. All isolates collected from 2016–2020 were confirmed as *M. paradoxa* through GBS data analysis.

### 2.2. GBS Sequencing and SNP Analysis

All 61 *M. paradoxa* isolates were genotyped using GBS analyzed using two pipelines. The first was the TASSEL V2 pipeline resulting in 3063 SNP variants. After filtering following the parameters described in the methods four individuals were removed and 952 variants were kept. The second pipeline used was GBS-SNP-CROP resulting in 15,098 SNP variants. After filtering, six isolates were removed and 4455 SNPs were retained.

Clonality was determined by comparing the identity by state (IBS) for the eight technical replicates to determine identity cutoffs for clonal genotypes. In the TASSEL generated variants, the IBS of technical replicates ranged from 0.88 to 0.92. An IBS clonality cutoff of 0.88 was applied to the entire TASSEL generated dataset. Clonal analysis showed that all isolates sequenced belong to one clonal lineage. In the GBS-SNP-CROP variants, IBS for the four technical replicates ranged from 0.93 to 0.85. An IBS clonality cutoff of 0.84 was applied to the entire GBS-SNP-CROP generated dataset. Clonal analysis revealed that all isolates sequenced using both variant discovery methods belong to one clonal lineage.

### 2.3. Field Evaluation of Leaf Rust and Stem Canker

Field ratings were collected in July of 2020 for leaf rust and stem canker in the 2013 Family Selection Trial (Figure 1a). Two families, *S. purpurea* × *S. suchowensis* and *S. koriyanagi* × *S. suchowensis*, displayed leaf rust and stem canker symptoms in the field (Table 1). We observed greater rust severity in the *S. purpurea* × *S. suchowensis* family as many plots were completely defoliated by leaf rust. Not all shrubs with stem cankers also had leaf rust, however high leaf rust disease severity was frequently associated with canker presence (Figure 1b). Stem cankers were commonly observed at stem nodes or friction points between shoots (Figure 1c). Cankers were black and necrotic and commonly displayed splitting of stems. Cankers frequently displayed bright yellow-orange rust uredospore pustules at the margins. Cankers were brought into the lab for isolation of potential pathogens. Five separate fungal cultures isolated from separate stems shared similar morphology when grown in culture. Using a compound microscope, conidia were observed to be oval to fusiform in shape, and mycelia were colored light grey to pink (Figure 2). The amplified PCR products of the ITS region from these cultured fungi were sequenced and the resulting reads were highly similar (100% pairwise identity) to the ITS sequence of *Colletotrichum salicis* (GenBank: MT068551.1). The ITS amplicons from other fungi isolated from cankers were sequenced and were highly similar to ITS sequences of species in the family Didymellaceae (GenBank: MT453292, 100% pairwise identity), *Fusarium sportrichioides* (GenBank: MT957569, 100% pairwise identity), and *Alternaria* spp. (GenBank: MW486028, 100% pairwise identity). Additionally, DNA from leaf and stem rust collected in 2020 was extracted and sequencing of ITS amplicons was performed as above. Resulting sequences were highly similar to known ITS sequences of *M. paradoxa* (Appendix A).

### 2.4. Host Resistance Mapping

QTL were identified for both leaf rust severity and rust canker presence, one for each of the parental backcross linkage maps of the *S. purpurea* × *S. suchowensis* F_1_ family (Figure 3). The QTL for leaf rust severity was found on chromosome 1 of the female *S. purpurea* parent, clone ID 94006. This QTL spanned from 169–175 cm, equating to a physical distance of 0.033 Mb (3.495–3.528 Mb) (Figure 3a). The peak marker, S01_3499289, was correlated with increased susceptibility to *M. paradoxa* in individuals with the heterozygous allele contributed by 94006 and accounted for 29% of the total phenotypic variation (Figure 3c). The QTL for stem rust canker presence was mapped to the male *S. suchowensis* parent, P63, and spanned from 15–19 cm (2.74–2.81 Mb) on chromosome 19, a distance of 0.7 Mb (Figure 3b). This peak marker, S19_2783471, explained 18.3% of the variation (Figure 3d).

### 2.5. Recovery of *M. paradoxa* from Dormant *Salix* Stems after Overwintering

Cuttings from dormant shoots containing stem cankers with no observable rust uredospore pustules were collected in mid-winter in January from eight different *S. purpurea* × *S. suchowensis* hybrid progeny. These cuttings were planted in potting mix and placed in the greenhouse. After approximately three weeks, four cuttings produced observable stem uredospore pustules (Figure 4). These stem uredospores were transferred to *S. purpurea* ‘Fish Creek’ detached leaves and disease was observed on these leaves after 10 d. Stem isolations were performed and *C. salicis* was identified from these overwintering cankers as above. Additionally, extracted DNA was submitted for GBS and clonal lineage analysis revealed that rust emerging from overwintered stem cankers was clonal with previously collected rust isolates. 

## 3. Discussion

This study utilized both genotypic and phenotypic data to describe the *M. paradoxa* isolates infecting three neighboring fields in Geneva NY over five growing seasons. These data show that all *M. paradoxa* isolates sampled across these years are from the same asexual clonal lineage. Both SNP data analysis methods, one with the use of a non-species reference genome and the other using a mock reference genome, yielded a set of differential SNPs. When IBS was analyzed using either method, the genotypic diversity index of all isolates was lower than the clonality threshold, indicating that all isolates collected in this study were a single clonal genotype regardless of year of collection. It is important to note, however that the IBS index cutoff threshold determined here was lower than most other GBS population biology studies, which are typically around 0.95 [8,13]. This is likely due to the use of a non-species reference genome resulting in poor alignment of fungal reads and poor SNP calling [14]. The percent identity of ITS region sequences between these two *Melampsora* species was 90.8% [6]. When we compared the SNP profiles of the complete TASSEL SNP dataset after filtering of representative *M. paradoxa* and *M. americana* isolates, the percent identity was only 37.6% [8]. This represents a very low level of similarity and supports the hypothesis that aligning *M. paradoxa* GBS reads to the *M. americana* reference genome lowered the SNP call quality. The other SNP analysis method using GBS-SNP-CROP identified a greater number of SNPs, however IBS was similarly low and was comparable to the approximate 84% accuracy described by the authors of this software [15].

These data suggest that asexual uredospores dominate the disease spread within a single field and that no sexual recombination occurred between growing seasons for the isolates collected. This does not conform with the life cycle described for *M. paradoxa*, in which alternation and sexual recombination on a *Larix* spp. host was presumed to be essential [7]. Additionally, observations in the field in 2020 provide evidence of stem cankers from which uredospore pustules develop around the margin. We hypothesize that this is the mechanism of asexual overwintering, as other *Melampsora* spp. rusts in Europe and Asia have been shown to asexually overwinter in stem cankers [16,17]. Additionally, uredospore stem pustules developed on the dormant stems brought into the greenhouse after overwintering in the field through January 2020, providing direct evidence of uredospore stage survival in harsh winter temperatures. From 1 December 2019 through January 2020, the mean minimum daily air temperature recorded at a weather station within 1 km of the site was −4.3 °C with a low air temperature of −14.5 °C and only 11 days in which the air temperature did not go below 0 °C (weather data available at http://NEWA.cornell.edu/ (accessed on 6 September 2022)). While the presence of stem cankers was not assessed in these fields between 2015 to 2018, it is likely that stem canker formation was the mechanism of overwintering for those isolates as well. It is not known if this population of *M. paradoxa* is unique and that most other *M. paradoxa* genotypes sexually recombine on an alternate host, however these data confirm that *M. paradoxa* is capable of an autoecious life cycle overwintering asexually.

The sampling of *M. paradoxa* coincided with a larger study collecting and describing the *M. americana* population in the northeast and Great Lakes region of the US [8]. Previous work had identified that *M. americana* would likely play the largest role in disease observed on *S. purpurea*, and the data presented here support that hypothesis [6]. *Melampsora paradoxa* isolates were only collected in three neighboring breeding fields at one site on the Crittenden North farm in Geneva NY. This contrasts with *M. americana* where isolates were collected across the northeast and Great Lakes region uniformly. Indeed, hosts from fields utilized in this study including the *S. purpurea* F_2_ mapping population and *Salix* F_1_ HCP where predominantly infested with *M. americana* rust isolates. However, an interesting shift in population occurred between the years of 2017 and 2018. In 2017, 62% of isolates collected from the *Salix* F_1_ HCP were *M. americana*, but in 2018 every isolate (*n* = 26) collected in this population was *M. paradoxa*. One possible mechanism of this shift was the coppicing of this trial that occurred during the winter of 2018. Perhaps the coppicing event allowed for the *M. paradoxa* isolates overwintering in the cutback stems to infect the newly growing host tissue in the early summer of 2018 and establish heavy disease before the *M. americana* spores traveled from the alternate hosts. Alternatively, there may be specific environmental conditions that favored *M. paradoxa* infection over *M. americana* infection in 2018 that we do not yet understand.

*Melampsora paradoxa* isolates were collected primarily from the 2013 Family Selection Trial in the years 2015 to 2017 and the *Salix* F_1_ HCP in 2018. The host species from which *M. paradoxa* was collected include *S. miyabeana,* and hybrid progeny of *S. purpurea* × *S. suchowensis*, *S. purpurea* × *S. udensis* and *S. purpurea* × *S. koriyanagi.* Additionally in the 2020 leaf rust field ratings, the *S. koriyanagi* × *S. suchowensis* hybrid progeny was observed to be susceptible to this pathogen. Together this represents quite a diverse range of host genotypes that are susceptible to this clonal lineage of *M. paradoxa*, while previous work identified *M. paradoxa* infecting primarily *S. eriocephala*, *S. nigra,* and *S. amygdaloides* in natural stands and in short-rotation coppice plantings [7]. This is particularly interesting since previous work showed that *M. paradoxa* was unable to infect most hosts assayed, representing six different host species [6]. Since most of the hosts that showed disease in the field were members of segregating breeding populations, it’s likely that the segregation of resistance genes provided an opportunity for *M. paradoxa* to spread over time. In fact, the two families in the 2013 Family Selection Trial that showed high levels of disease share a male parent, *S. suchowensis* P63, that previously was found to be resistant to a *M. paradoxa* isolate that was clonal to all other *M. paradoxa* isolates collected in this project [6]. This might suggest that this *S. suchowensis* host contains resistance genes that are segregating in its hybrid progeny families. However, it’s the allele from *S. suchowensis* P63 on chromosome 19, identified through QTL mapping, that is partially correlated with stem canker presence in the F_1_ progeny.

In combination with leaf rust, black stem cankers were frequently observed in conjunction with stem uredospore pustules. They were typically found at friction points between willow shoots or at node junctions. Additionally, stem uredospore pustules were always observed in conjunction black cankers, however all black cankers were not always observed with rust uredospores. This led to the hypothesis that other cankering fungi may play a role in this disease. Willow is known to be susceptible to several cankering fungi, specifically *C. salicis*, *Venturia saliciperda*, and *Botryosphaeria dothidea* [3]. Of the culturable fungal isolates from cankers in this study, *C. salicis* was repeatedly isolated and identified based on the ITS region DNA sequence. *Colletotricum salicis* (formerly *Glomerella miyabeana*) is known to cause black canker disease of willow and can result in complete shoot death from girdling of the stem vasculature [18]. It is intriguing that this cankering fungus was isolated in conjunction with *M. paradoxa*. To date there is no literature that described a requirement for *C. salicis* in rust stem canker formation, but in this study *M. paradoxa* was frequently observed on stems with cankers from which *C. salicis* was isolated. This repeated observation might suggest an interaction between these fungi. Additionally, *M. americana* was never isolated from any stem cankers, even though this fungus was the most prevalent species until 2018. This provides an exciting avenue for future research dissecting the exact relationship between these fungi.

This study tracked *M. paradoxa* in Geneva, NY over five growing seasons in three fields. These were the only three fields in which *M. paradoxa* was identified in the northeast US during this time period. Asexual overwintering and stem uredospore pustules associated with stem cankers were observed both in the field and in the greenhouse when dormant stems were brought in from the field. Additionally, *C. salicis* was identified and associated with these cankers. This is the first extensive study exploring and providing evidence for asexual overwintering of *M. paradoxa* in North America. Future work could explore the exact relationship between *M. paradoxa* and the isolated culturable fungi through co-inoculation experiments and field overwintering experiments. This work highlights the potential threat to *Salix* bioenergy crops posed by asexual reproduction of this species.

## 4. Materials and Methods

### 4.1. *Melampsora paradoxa* Isolate Collection and GBS Genotyping

A total of 61 *M. paradoxa* isolates were collected from cultivated shrub willow breeding populations on the Crittenden North farm at Cornell AgriTech in Geneva, NY in neighboring research fields with multiple hybrid pedigrees in the summers of 2015, 2016, 2017 and 2018 (Table 1) [11,19]. Ten symptomatic rust-infected leaves per shrub were collected in the field, stored in plastic bags in a cooler, and returned to the lab for single pustule isolations. The following steps were conducted as previously described [6]. Three uredospore pustules from each set of 10 leaves were used to generate three separate single-genotypic isolates by performing three successive rounds of single spore inoculations. Isolates were maintained on *S. purpurea* ‘Fish Creek’ in water agar Petri dishes. All instruments used in isolation and collection of rust uredospores were sterilized between isolates using 70% ethanol. Following single pustule isolations, each individual isolate was grown on approximately 10 detached *S. purpurea* leaves, uredospores were collected using a cyclone spore collector, and DNA was extracted. Rust species identification was determined following GBS sequencing and comparison to known *M. americana* and *M. paradoxa* SNP profiles following methods described by Crowell et al. [8]. Species type isolates used in this GBS sequencing comparison were known from ITS sequencing and phylogenetic analysis [6]. Samples containing 25 µL of genomic DNA (≥20 ng µL^−1^) per isolate were submitted to University of Wisconsin-Madison Biotechnology Center for GBS library preparation and next-generation sequencing. Isolates collected from 2015 and 2016 were sequenced using Illumina HiSeq2500 yielding single-end 100 bp reads and isolated collected from 2017 and 2018 were sequenced using NovaSeq6000 generating paired-end 150 bp reads. This change in sequencing method reflected changes within the Biotechnology Center’s sequencing infrastructure that occurred between plate submissions.

### 4.2. SNP Analysis

Variant discovery in GBS data was completed using two separate methods. The first was a non-species reference SNP calling method using the *M. americana* reference genome R15-033-03 (https://mycocosm.jgi.doe.gov/Melame1/Melame1.home.html (accessed on 9 August 2022)) and the TASSEL 5 V2 pipeline [20]. The second method was a mock referenced SNP calling method using GBS-SNP-CROP [15]. For both methods, only the forward reads of the submitted plates of sequencing were used. Scripts and data related to this analysis are available in the GitHub repository: https://github.com/crc255/M-paradoxa_population_biology_data (accessed on 6 September 2022).

TASSEL 5 V2 Methods: Barcode sequences were trimmed, and genotype calling was performed using the TASSEL GBSv2 pipeline following default parameters [20,21]. GBS reads were aligned to the *M. americana* R15-033-03 reference genome [8] for variant discovery using the Burrows-Wheeler algorithm bwa-aln v0.7.17 with default parameters [22]. TASSEL 5 was used to remove individuals with >40% missing data.

GBS-SNP-CROP Methods: Raw reads were parsed, barcodes were trimmed using Trimmomatic v0.39 [23], and sequences were demultiplexed using the GBS-SNP-CROP demultiplexer. Fifteen demultiplexed samples with the greatest number of reads were used to generate a mock reference using VSEARCH v2.15 [24]. Reads were aligned to the mock reference using the Burrows-Wheeler algorithm bwa-aln v0.7.17 with default parameters and the remaining steps of the GBS-SNP-CROP pipeline were completed with default parameters [22].

The resulting variant files from both pipelines were filtered using the following parameters in Vcftools v0.1.17 [25]: genotypes with <5× coverage were set to missing; minor allele frequency (MAF) cutoff was set to ≥0.05; Missing data cutoff was <40%; and indels were removed. The resulting vcf dataset was reexamined in TASSEL 5 and individuals with >40% missing data were filtered again.

### 4.3. Determination of Clonality

Clonal analysis was conducted using R v4.0.3 [26] in RStudio v1.4.1103 [27] and pair-wise IBS was determined as previously described [8]. Eight technical replicates, two from 2015, and six from 2018 were included to approximate sequencing error rate by comparing SNP profiles. This approximation was used to establish separate percent identity cutoffs for clonal lineages in both the TASSEL and GBS-SNP-CROP derived SNP dataset. Once this cutoff was determined, IBS was calculated, defined as the proportion of alleles shared at non missing sites, using previously described code [13]. Resulting IBS calculations were used to assign clonal lineages using the IBS of technical replicates as cutoffs.

### 4.4. Leaf Rust and Stem Canker Rust Disease Evaluation

Leaf rust and stem canker rust symptoms were observed on 5 June 2020 in the 2013 Family Selection Trial in Geneva, NY [28]. This field contained a total of 1136 plots with four replicates of selected clones from eight breeding families planted in randomized complete block design (five intraspecific hybrids and three *S. miyabeana* pedigrees plus commercial and pre-commercial cultivars). Border rows planted on both east and west sides of the field were comprised of replicate nursery bed plantings of each genotype planted in the trial. This field was rated for leaf rust severity as previously described [11] and stem canker incidence using a presence or absence (0 or 1) binary scale. Field ratings are available at: https://github.com/crc255/M-paradoxa_population_biology_data (accessed on 6 September 2022). Cuttings of infected stem tissue were returned to the lab for fungal isolation. Approximately 2 mm discs were excised from the margins of stem cankers, surface sterilized in a 10% bleach solution, and plated on acidified PDA for isolation of culturable fungi. DNA was extracted from resulting fungi using the Extact-N-Amp™ DNA isolation kit (Millipore-Sigma, Burlington, MA, USA). The ITS region of the cultured fungi from the stem canker was amplified by polymerase chain reaction (PCR) using ITS primers ITS4 and ITS5 [29]. Resulting PCR products were sequenced using Sanger sequencing at the Cornell University Institute of Biotechnology. Resulting DNA sequences of the culturable rust fungi were cross-referenced to the NCBI nucleotide database using the “blastn” function [30].

Rust was collected from both leaf tissue and stem cankers from the 2013 Family Selection Trial in 2020. Rust was inoculated onto detached leaves, DNA was extracted, and complete ITS region was amplified and sequenced as previously described [6]. Sequences of the rust ITS amplicons were aligned to sequences of known *Melampsora* type species for species identification (Appendix A) [6].

### 4.5. Host Resistance QTL Mapping

QTL mapping of host resistance in the *S. purpurea* × *S. suchowensis* F_1_ family used previously described parental, backcross linkage maps [12]. Following a significant Shapiro-Wilk test (*p* < 0.05), leaf rust severity was normalized using a Box-Cox transformation [31]. For both parental maps, QTL were identified and refined using the mean of the F_1_ individual in R/qtl [32]. Genotype probabilities were calculated using ‘calc.genoprob’ (step = 5, error.prob = 0.01, map.function = “kosambi”). Composite interval mapping via ‘cim’ for leaf rust severity and ‘scanone’ for canker presence was used to determine marker LOD scores, while 1000 permutation tests were completed to set the 0.1 genome-wide significance thresholds. Upon identifying a significant LOD score, the position of the peak marker and the percent variation explained were calculated using ‘refineqtl’ and ‘fitqtl’. The 1.5 LOD support intervals were not calculated in this study as the bounds of the QTL were less than the width of the interval.

### 4.6. Recovery of *M. paradoxa* from Dormant *Salix* Stems after Overwintering

Dormant stem cuttings (approximately 20 cm in length) containing cankers with no observable rust sporulation were collected in the field from individuals of the *S. purpurea* × *S. suchowensis* F_1_ family in the 2013 Family Selection Trial in Geneva, NY in February 2021. Cuttings were planted in potting mix and placed in the greenhouse with a day: night photoperiod of 14:10 h and max daytime and nighttime temperatures of 26 °C and 18 °C respectively. After approximately 3 weeks, disease symptoms were visually assessed, and rust was isolated on detached leaves of susceptible cultivar *S. purpurea* ‘Fish Creek’ in water agar Petri dishes as previously described [6]. Culturable fungi were isolated from the stem cutting cankers and both the rust and culturable fungi from the cankers were identified via sequencing ITS amplicons as described above.

## Figures and Tables

**Figure 1 plants-11-02385-f001:**
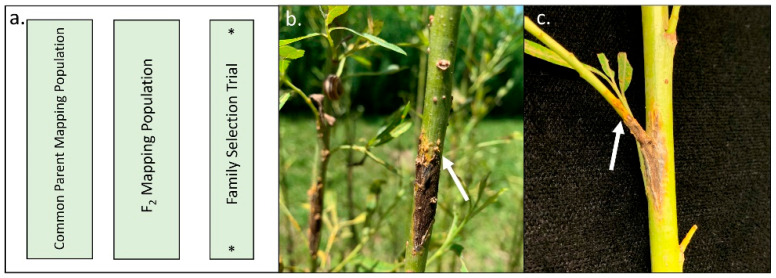
Trial proximity and photos of stem cankers observed in 2020. (**a**) Schematic of three neighboring fields for which *M. paradoxa* rust was collected, (*) indicates field where cankers were observed. (**b**) Image of stem canker on *Salix purpurea* × *S. suchowensis* shoot in 2013 Family Selection Trial. Image of stem cutting containing a canker lesion collected from the field. (**c**) Image of stem cutting containing a canker lesion collected from the field. Arrows indicate stem rust uredospore pustules at the margins of lesions.

**Figure 2 plants-11-02385-f002:**
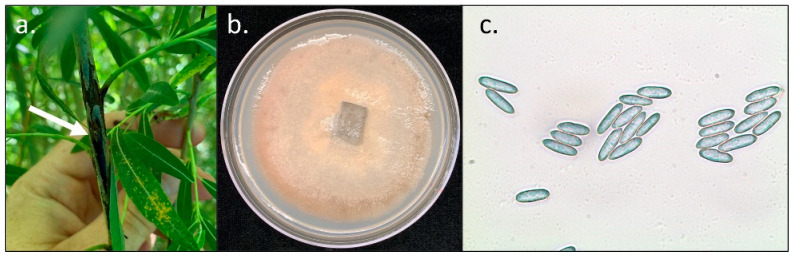
Photo of stem cankers without uredospore sporulation and fungal isolate of *C. salicis*. (**a**) Image of black canker in the 2013 Family Selection Trial (**b**) Image of isolated *Colletotrichum salicis* on potato dextrose agar. (**c**) Micrograph of *C. salicis* spores stained with cotton blue taken at 40× with a compound microscope.

**Figure 3 plants-11-02385-f003:**
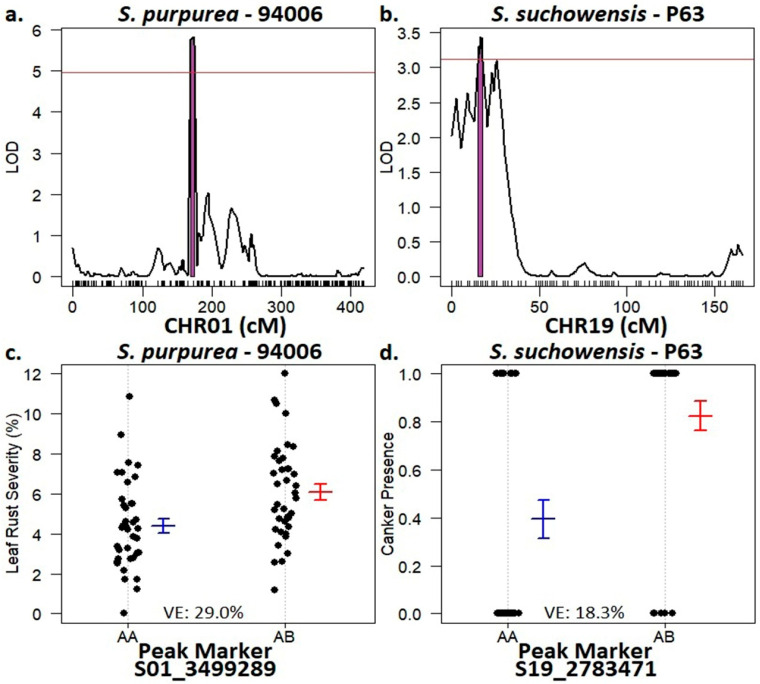
QTL results for leaf rust severity and stem canker presence in the *S. purpurea* × *S. suchowensis* F_1_ family. (**a**) Leaf rust severity QTL on *S. purpurea* 94006, CHR01. (**b**) Stem canker QTL on *S. suchowensis* P63, CHR19. (Panels (**a**) and (**b**)) The red line indicates the 1000-permutation genome wide significance threshold while the purple area shows the region of the QTL. (**c**) Zygosity of the peak marker, S01_3499289 from CHR01, associated with leaf rust severity in relation to genotype means. (**d**) Zygosity of the peak marker, S19_2783471 from CHR19, associated with stem canker presence in relation to genotype means. (Panels (**c**) and (**d**)) Blank dots show individual genotype means, bar and whiskers are the mean and standard error of each marker type.

**Figure 4 plants-11-02385-f004:**
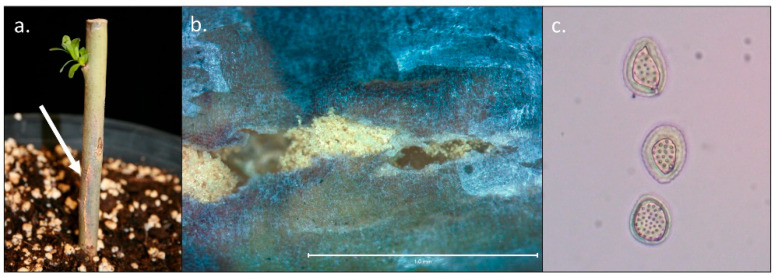
*Melampsora paradoxa* recovery from dormant stem canker after overwintering. (**a**) Image of cutting after approximately three weeks in the greenhouse following dormancy in the field. Arrow points to stem uredospore pustules that developed after stems broke dormancy (**b**) Micrograph of stem rust identified by arrow taken at 30× with dissecting microscope. (**c**) Micrograph of uredospores collected from stem pustules taken at 40× with a compound microscope.

**Table 1 plants-11-02385-t001:** *Melampsora paradoxa* isolate collection.

Rust ID	Year	Collection Date	Field	Host Pedigree	Clone/Cultivar
R15-001-01	2015	10 July 2015	Family Selection Trial	*S. purpurea* × *S. suchowensis*	10X-400-051
R15-002-01	2015	10 July 2015	Family Selection Trial	*S. purpurea* × *S. suchowensis*	10X-400-051
R15-003-01	2015	20 July 2015	Family Selection Trial	*S. purpurea* × *S. suchowensis*	10X-400-051
R15-004-01	2015	20 July 2015	Family Selection Trial	*S. purpurea* × *S. suchowensis*	10X-400-051
R15-005-01	2015	30 July 2015	Family Selection Trial	*S. purpurea* × *S. suchowensis*	10X-400-051
R15-006-01	2015	30 July 2015	Family Selection Trial	*S. purpurea* × *S. suchowensis*	10X-400-051
R15-007-01	2015	31 July 2015	Family Selection Trial	*S. purpurea* × *S. suchowensis*	10X-400-051
R15-008-01	2015	1 August 2015	Family Selection Trial	*S. purpurea* × *S. suchowensis*	10X-400-051
R15-009-01	2015	12 August 2015	Family Selection Trial	*S. purpurea* × *S. suchowensis*	10X-400-051
R15-010-01	2015	13 August 2015	Family Selection Trial	*S. purpurea* × *S. suchowensis*	10X-400-051
R16-003-03	2016	1 June 2016	Family Selection Trial	*S. purpurea* × *S. suchowensis*	10X-400-048
R16-004-01	2016	1 June 2016	Family Selection Trial	*S. purpurea* × *S. suchowensis*	10X-400-051
R16-005-01	2016	1 June 2016	Family Selection Trial	*S. purpurea* × *S. suchowensis*	10X-400-053
R16-006-01	2016	1 June 2016	Family Selection Trial	*S. purpurea* × *S. suchowensis*	10X-400-055
R17-001-02	2017	25 July 2017	Family Selection Trial	*S. purpurea* × *S. suchowensis*	10X-400-051
R17-002-01	2017	25 July 2017	Family Selection Trial	*S. purpurea* × *S. suchowensis*	10X-400-051
R17-002-02	2017	25 July 2017	Family Selection Trial	*S. purpurea* × *S. suchowensis*	10X-400-051
R17-002-03	2017	25 July 2017	Family Selection Trial	*S. purpurea* × *S. suchowensis*	10X-400-051
R17-003-01	2017	25 July 2017	Family Selection Trial	*S. purpurea* × *S. suchowensis*	10X-400-054
R17-004-01	2017	25 July 2017	Family Selection Trial	*S. purpurea* × *S. suchowensis*	10X-400-076
R17-004-03	2017	25 July 2017	Family Selection Trial	*S. purpurea* × *S. suchowensis*	10X-400-076
R17-005-01	2017	25 July 2017	Family Selection Trial	*S. purpurea* × *S. suchowensis*	10X-400-063
R17-007-03	2017	25 July 2017	Family Selection Trial	*S. purpurea* × *S. suchowensis*	10X-400-059
R17-008-02	2017	25 July 2017	*Salix* F_1_ HCP *	*S. purpurea* × *S. koriyanagi*	13X-438-045
R18-002-01	2018	14 June 2018	Family Selection Trial	*S. purpurea* × *S. suchowensis*	10X-400-094
R18-003-02	2018	14 June 2018	Family Selection Trial	*S. miyabeana*	01-200-007
R18-003-03	2018	14 June 2018	Family Selection Trial	*S. miyabeana*	01-200-007
R18-004-02	2018	21 June 2018	*Salix* F_1_ HCP	*S. purpurea* × *S. udensis*	13X-358-003
R18-004-03	2018	21 June 2018	*Salix* F_1_ HCP	*S. purpurea* × *S. udensis*	13X-358-003
R18-005-03	2018	21 June 2018	*Salix* F_1_ HCP	*S. purpurea* × *S. udensis*	13X-358-001
R18-006-01	2018	21 June 2018	*Salix* F_1_ HCP	*S. purpurea* × *S. udensis*	13X-358-157
R18-006-03	2018	21 June 2018	*Salix* F_1_ HCP	*S. purpurea* × *S. udensis*	13X-358-157
R18-007-02	2018	21 June 2018	*Salix* F_1_ HCP	*S. purpurea* × *S. udensis*	13X-358-124
R18-007-03	2018	21 June 2018	*Salix* F_1_ HCP	*S. purpurea* × *S. udensis*	13X-358-124
R18-008-01	2018	21 June 2018	*Salix* F_1_ HCP	*S. purpurea* × *S. udensis*	13X-358-177
R18-009-02	2018	21 June 2018	*Salix* F_1_ HCP	*S. purpurea* × *S. koriyanagi*	13X-438-027
R18-010-01	2018	21 June 2018	*Salix* F_1_ HCP	*S. purpurea* × *S. udensis*	13X-358-003
R18-010-02	2018	21 June 2018	*Salix* F_1_ HCP	*S. purpurea* × *S. udensis*	13X-358-003
R18-010-03	2018	21 June 2018	*Salix* F_1_ HCP	*S. purpurea* × *S. udensis*	13X-358-003
R18-013-01	2018	25 June 2018	*Salix* F_1_ HCP	*S. purpurea* × *S. suchowensis*	13X-443-020
R18-013-03	2018	25 June 2018	*Salix* F_1_ HCP	*S. purpurea* × *S. suchowensis*	13X-443-020
R18-014-01	2018	25 June 2018	*Salix* F_1_ HCP	*S. purpurea* × *S. udensis*	13X-358-171
R18-015-01	2018	25 June 2018	*Salix* F_1_ HCP	*S. purpurea* × *S. suchowensis*	13X-440-148
R18-017-01	2018	25 June 2018	*Salix* F_1_ HCP	*S. purpurea* × *S. udensis*	13X-358-192
R18-018-02	2018	25 June 2018	*Salix* F_1_ HCP	*S. purpurea* × *S. udensis*	13X-358-118
R18-019-01	2018	25 June 2018	*Salix* F_1_ HCP	*S. purpurea* × *S. suchowensis*	13X-440-144
R18-019-02	2018	25 June 2018	*Salix* F_1_ HCP	*S. purpurea* × *S. suchowensis*	13X-440-144
R18-020-01	2018	25 June 2018	*Salix* F_1_ HCP	*S. purpurea* × *S. koriyanagi*	13X-438-094
R18-022-03	2018	28 June 2018	*Salix* F_1_ HCP	*S. purpurea* × *S. udensis*	13X-358-013
R18-023-03	2018	28 June 2018	*Salix* F_1_ HCP	*S. purpurea* × *S. suchowensis*	13X-443-031
R18-024-01	2018	28 June 2018	*Salix* F_1_ HCP	*S. purpurea* × *S. udensis*	13X-358-135
R18-024-02	2018	28 June 2018	*Salix* F_1_ HCP	*S. purpurea* × *S. udensis*	13X-358-135
R18-027-03	2018	28 June 2018	*Salix* F_1_ HCP	*S. purpurea* × *S. udensis*	13X-358-085
R20-002	2020	8 June 2020	Family Selection Trial	*S. purpurea* × *S. suchowensis*	10X-400-043
R20-003	2020	8 June 2020	Family Selection Trial	*S. purpurea* × *S. suchowensis*	10X-400-053
R20-027	2020	8 June 2020	Family Selection Trial	*S. purpurea* × *S. sociogenesis*	10X-400-051
R20-027(S)	2020	8 June 2020	Family Selection Trial	*S. purpurea* × *S. suchowensis*	10X-400-051
R20-047	2020	13 June 2020	Family Selection Trial	*S. purpurea* × *S. suchowensis*	10X-400-057
Overwinter-Canker1	2020	10 February 2021	Family Selection Trial	*S. purpurea* × *S. suchowensis*	10X-400-009
Overwinter-Canker2	2020	10 February 2021	Family Selection Trial	*S. purpurea* × *S. suchowensis*	10X-400-009

* HCP, Hybrid Common Parent Population.

## Data Availability

Data are available in the Appendix A associated with this article and at the public data repositories cited. Any additional information needed is available upon request.

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
