# Peer review of "Evidence of Asexual Overwintering of Melampsora paradoxa and Mapping of Stem Rust Host Resistance in Salix"

_plants, 2022, doi:10.3390/plants11182385_

Round 1

Reviewer 1 Report

Minor comments/suggestions: 

Results:

Figure 1: suggest that the authors move “(*) indicate field where cankers were observed” to the (a) part of the description.

“The amplified PCR products of the ITS region from cultured fungi were sequenced and resulting reads were highly similar to Colletotrichum” – can the authors give the percentages here and in the following sentences?

Discussion: Do the authors have access to temperature changes in Geneva, NY and could add that to the discussion? Specifically “uredospore stem pustules developed on the dormant stems brought into the greenhouse after overwintering, providing direct evidence of uredospore stage survival in harsh winter temps.”

Methods: There are a few instances in 4.1 where S. purpurea, M. americana, and M. paradoxa are not italicized.

Author Response

Thank you to the reviewer for your close attention to detail.  Please find a list of Reviewer 1 comments and our response to all below.  All edits to the manuscript are indicated in the revised manuscript. 

Reviewer 1

Figure 1: suggest that the authors move “(*) indicate field where cankers were observed” to the (a) part of the description.

The suggested formatting was incorporated into the Figure 1 legend. 

“The amplified PCR products of the ITS region from cultured fungi were sequenced and resulting reads were highly similar to Colletotrichum” – can the authors give the percentages here and in the following sentences?

Thanks for this suggestion - the pairwise percent identity scores for each BLAST search were added for the culturable fungi identified in the results section.  

Discussion: Do the authors have access to temperature changes in Geneva, NY and could add that to the discussion? Specifically “uredospore stem pustules developed on the dormant stems brought into the greenhouse after overwintering, providing direct evidence of uredospore stage survival in harsh winter temps.”

Thanks for this suggestion - we added information into the discussion section on low air temperatures recorded at a weather station within 1 km of the site for the period before collection.  

Methods: There are a few instances in 4.1 where S. purpureaM. americana, and M. paradoxa are not italicized.

Thanks for catching these - we apologize for these errors. All italics errors were identified and corrected.  

Reviewer 2 Report

The paper provides evidence for asexual overwintering of Melampsora paradoxa in North America. Stem uredospore pustules associated with stem cankers were observed on shrub willow. Additionally, host resistance in Salix was identified on chromosomes 1 and 19 through QTL mapping.

In general the paper is well written and of a high standard. I would like to make some minor critical comments:

The Results section is not clearly separated from the methodological part. Here it needs to place more emphasis on presenting only the results. On the other hand, methodological procedures of GBS and QTL analysis sometimes are not presented clearly for readers not familiar with this scientific field. For example: width of the interval – width of the confidence interval, etc.

Some M. paradoxa ITS sequences in Figure S1 belong to isolates not found in Table 1: R15-001_ITS, R15-003_ITS, R15-004_ITS R15-005_ITS.

spp – spp.

very a low – a very low

tassel – Tassel

though co-inoculation – through co-inoculation

ExtactnAmp – ExtractnAmp

MilliporeSigma – Millipore-Sigma

Author Response

Thank you to the reviewer for your close attention to detail.  Please find a list of the Reviewer 2 comments and our responses to all comments. All edits are indicated in the revised manuscript. 

Reviewer 2

The Results section is not clearly separated from the methodological part. Here it needs to place more emphasis on presenting only the results. On the other hand, methodological procedures of GBS and QTL analysis sometimes are not presented clearly for readers not familiar with this scientific field. For example: width of the interval – width of the confidence interval, etc.

Thank you for identifying potential areas of confusion.  Sections of results were modified to more effectively differentiate them from methods.  Additionally, sections of methods were modified to clarify points of confusion. 

Some M. paradoxa ITS sequences in Figure S1 belong to isolates not found in Table 1: R15-001_ITS, R15-003_ITS, R15-004_ITS R15-005_ITS.

Thanks for catching these errors. These rust isolates were added to Table 1

Various text errors: spp – spp., very a low – a very low, tassel – Tassel, though co-inoculation – through co-inoculation, ExtactnAmp – ExtractnAmp, MilliporeSigma – Millipore-Sigma. 

Thanks for catching these errors.  All identified text errors were corrected.
